# STAN: A Spatio-Temporal Attention Network for Space Debris Multistage Collision Avoidance

## Abstract

The rapid expansion of space missions has led to an exponential increase in space debris, posing severe threats to spacecraft. Existing approaches struggle to handle multistage collision risks in cluttered orbital environments, and the use of continuous low-thrust propulsion further complicates avoidance planning. To address these challenges, we propose the **S**patio-**T**emporal **A**ttention **N**etwork (**STAN**), which employs novel **S**patio-**T**emporal **Attention** (**ST-Attention**) layers in place of conventional attention mechanisms. STAN encodes satellite-debris pairs and integrates time and distance into attention weight computation, enabling the model to generate context-aware low-thrust maneuvers. The model is trained using deep reinforcement learning across four representative multistage collision scenarios, jointly optimizing collision probability, fuel consumption, and orbital deviation. Experimental results show that STAN outperforms baseline methods in safety performance, fuel efficiency, and orbit preservation.

## 1 Introduction

The rapid advancement of space technologies in recent decades has led to an exponential increase in space activity. However, this progress leads to the explosive growth of orbital debris. As of 2025, over 100,000 pieces of debris larger than 1 cm orbit the Earth, posing significant threats to space missions (CelesTrak, 2025). Even tiny particles can cause catastrophic collisions, which can also generates the potential for **multistage collisions**, where secondary collisions arise as a result of earlier avoidance maneuvers (European Space Agency (ESA), 2016).

To mitigate these risks, the space community relies heavily on collision risk assessment and avoidance maneuver planning. Once a high-risk event is identified, an avoidance maneuver is executed—typically using either impulsive or continuous thrust strategies. While impulsive maneuvers are intense and discrete, they are unsuitable for spacecraft using electric propulsion systems. As a result, **continuous low-thrust maneuver planning** has received increasing attention in recent years. In parallel, research on autonomous collision avoidance using intelligent methods has started to gain traction. However, Current approaches face several key limitations. First, they are not suitable for multistage collisions because existing pairwise decomposition methods are computationally expensive and overlook debris coupling. Second, many policy architectures have a limited input capacity as they assume fixed input sizes, which restricts their ability to generalize to variable debris counts and large-scale threats. Finally, they have restricted control strategies and lack the fine-grained control needed for real-time, low-thrust maneuver planning.

To address these gaps, this work proposes **STAN (Spatio-Temporal Attention Network)**: a policy network designed for continuous low-thrust, real-time debris avoidance in complex orbital environments. As shown in Figure 1, STAN uses **ST-Attention (Spatio-Temporal Attention)**, enabling it to process arbitrary numbers of debris objects and prioritize threats based on both spatial proximity and temporal collision risk, more details will be introduced in section 4 . The system uses a two-body dynamic model for state propagation, uses screening methods to obtain high-risk debris, and applies a PPO-based deep reinforcement learning algorithm to train the maneuver policy end-to-end. Our contributions can be summarized as follows:

1. Introduce a physics-inspired **spatio-temporal attention mechanism (ST-Attention)** for collision avoidance. Ablation studies demonstrate accelerated initial learning and a strong dependency on the physical bias in the later stages of training.

2. Develop the **STAN architecture** that supports **variable-length input**, achieving up to 81% fuel savings and 746% higher rewards than baselines, while keeping the collision probability below 0.002.

3. Evaluate STAN on several **large-scale scenarios** with up to 116 debris. Results demonstrate competitive performance in both maneuver efficiency and generalization.

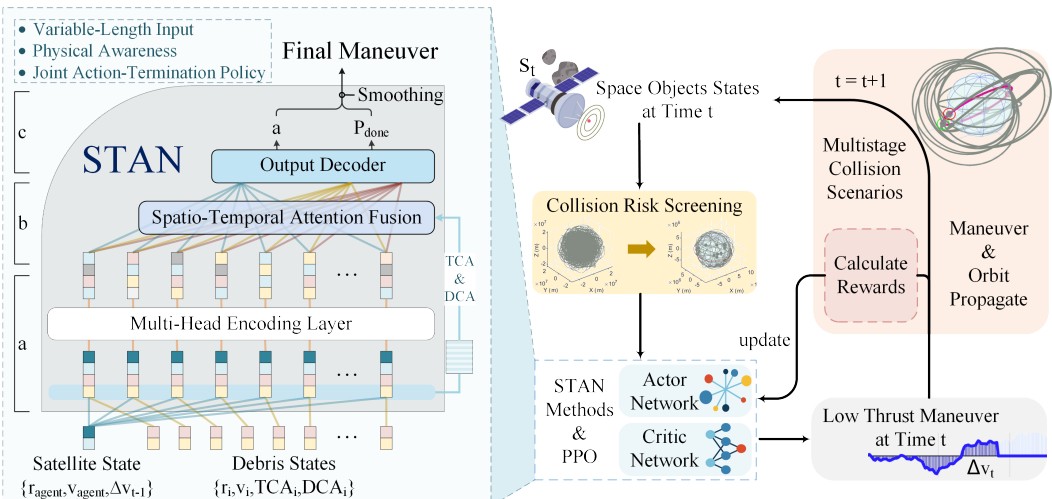

Figure 1: Diagram of STAN methods. The system processes space objects' states through a risk screening module. The risky targets are then fed into the STAN network for real-time maneuver outputs. These maneuvers affect the environment, and the resulting rewards are used to optimize using PPO algorithm. The detailed structure of the STAN architecture is shown on the left.

## 2 RELATED WORK

The related work is presented from three perspectives: early warning of space debris collisions, optimization-based avoidance for debris collision and intelligent avoidance for debris collision.

### 2.1 EARLY WARNING OF SPACE DEBRIS COLLISIONS

Spacecraft collision avoidance relies on accurate orbit prediction and collision probability assessment. Patera proposed an encounter-frame-based approach for collision probability estimation (Patera, 2001). In parallel, orbit propagation methods such as SGP4/SDP4 became standard tools in debris prediction (Hoots et al., 2004; Vallado et al., 2006). Risk evaluation methods evolved from box-region approaches to probability-based approaches (Akella & Alfriend, 2000). Foster proposed polar integration method Foster & Estes (1992). Chan proposed analytical probability approximation (Chan, 1997). Bai considered uncertainty and established an orbit prediction model for risk estimation (Bai et al., 2013). Casanova et al. explored continuous-thrust collision avoidance through a triple-filter method (Casanova et al., 2014). Hongqiang et al. demonstrated that probabilistic methods offer better performance under onboard resource constraints (Hongqiang et al., 2020). Probabilistic methods significantly improved risk quantification accuracy.

### 2.2 OPTIMIZATION-BASED AVOIDANCE FOR DEBRIS COLLISION

Collision avoidance maneuver design has long been addressed through optimal control theory. Alfano proposed an impulsive avoidance strategy (Alfano, 2006). Mueller et al. modeled the planning as a nonlinear programming problem (Mueller, 2009; Mueller et al., 2013). Bombardelli and

Hernando-Ayuso extended the framework to low Earth orbit settings (Bombardelli & Hernando-Ayuso, 2015). For continuous thrust planning, Bombardelli and Hernando-Ayuso further studied the obstacle avoidance strategy under low-thrust conditions (Hernando-Ayuso & Bombardelli, 2021). However, these methods often suffer from limited adaptability in rapidly evolving space traffic environments, motivating the use of intelligent learning-based alternatives.

## 2.3 INTELLIGENT AVOIDANCE FOR DEBRIS COLLISIONS

Recent research has explored the potential of intelligent learning-based methods to enhance decision-making under uncertainty. Seong and Kim (2012) compared genetic algorithms, particle swarm optimization, and differential evolution for multi-threat avoidance (Seong & Kim, 2015). Early work by Kolosa demonstrated the feasibility of applying RL to low-thrust spacecraft control (Kolosa, 2019). Miller and Linares further demonstrated that RL-based low-thrust optimal control can match classical indirect methods while offering improved scalability in high-dimensional settings (Miller & Linares, 2019). Yuan and Li used a Markov Decision Process to model the spacecraft avoidance problem, capturing high-dimensional constraints (Yuan & Li, 2022). Qu et al. applied Deep Deterministic Policy Gradient (DDPG) with adaptive policy gradients to generate continuous low-thrust maneuvers (Qu et al., 2022). Liu et al. used Proximal Policy Optimization (PPO) to avoid debris under dynamic proximity threats (Liu et al., 2023). LaFarge et al. introduces a hybrid architecture, combining reinforcement learning (RL) and differential corrections, to rapidly generate low-thrust maneuvers (LaFarge et al., 2023). Kazemi et al. validates the use of a Proximal Policy Optimization approach for effective single-object maneuver planning to achieve collision avoidance (Kazemi et al., 2024). Mu proposed a safe RL framework based on PPO, incorporating LSTM to handle time-varying debris fields (Mu et al., 2024). For multi-agent obstacle avoidance scenarios Mu also proposed an orbital chase defense game model based on multi-agent proximal strategy optimization (Mu et al., 2025). Solomon and Paduraru (2024) provided a comparative analysis of REINFORCE, PPO, and DQN in low-thrust trajectory planning (Solomon & Paduraru, 2024). In parallel, recent studies have investigated attention-based architectures. Tao et al. proposed SDebrisNet, a spatial–temporal saliency network for debris detection in orbital imagery (Tao et al., 2023). In the broader domain of trajectory prediction, Shen et al. introduced an improved attention mechanism for human-like vehicle trajectory forecasting (Shen et al., 2023). Similarly, Jiang et al. proposed an attention-based deep reinforcement learning model for human-like collision avoidance in autonomous ships (Jiang et al., 2022). From a methodological perspective, Jin et al. provided a comprehensive survey of spatio-temporal graph neural networks and their variants (Jin et al., 2024), while subsequent creative works, such as the Equivariant Spatio-Temporal Attentive Graph Network further extend this line of research (Wu et al., 2023). Despite their contributions, these approaches often target perception or trajectory forecasting tasks outside the spacecraft domain, lack integration with reinforcement learning for continuous low-thrust control, and most importantly, have shown limited effectiveness in multistage collision scenarios.

## 3 PRELIMINARIES

### 3.1 DYNAMICS AND LOW-THRUST CONSTRAINTS

To simulate multi-stage collisions and formulate the reinforcement learning environment, space objects are represented in the inertial coordinate system. Since the aim of this study is to verify the feasibility of the method rather than to predict trajectories precisely, each object's motion is modeled as a two-body problem governed by Newtonian gravity. The governing second-order differential equation is:

$$\frac{d^2\mathbf{r}}{dt^2} = -\frac{\mu}{r^3}\mathbf{r} \tag{1}$$

where $\mathbf{r} \in \mathbb{R}^3$ is the position vector of the object relative to Earth, and $\mu = 398600.4418\,\text{km}^3/\text{s}^2$ is the standard gravitational parameter. To meet the characteristic of continuous small thrust, in the real-time control of each step, we limit the acceleration of the spacecraft to the maximum acceleration that the current electric thruster can achieve, taking $\|\frac{d^2\mathbf{r}}{dt^2}\| \leq 2.27 \times 10^{-4} m/s^2$.

### 3.2 EARLY WARNING METHODS FOR DEBRIS COLLISIONS

This work presents a two-stage framework for collision risk screening: a **static screening** based on orbital altitude, followed by a **dynamic screening** based on temporal proximity. The initial static step efficiently filters out debris that does not have an orbital overlap with the primary spacecraft. The subsequent dynamic step further eliminates objects where the difference in passage time through intersection points exceeds a predefined threshold. This methodology substantially reduces the number of potential threats, thereby enhancing the efficiency of subsequent risk assessment, learning, and planning. Please refer to the Appendix for a more detailed description.

### 3.3 MULTISTAGE COLLISION SCENARIO GENERATION

To support effective evaluation and training, a progressive set of multistage collision scenarios was designed to capture the dynamic and uncertain nature of orbital environments. These scenarios simulate conditions where a spacecraft, after avoiding an initial collision, faces subsequent threats from new debris. Figure 2 shows the four types of collision scenarios: **single debris collision, strict multistage collision, probability-based multistage collision** and **complex multistage collision**.

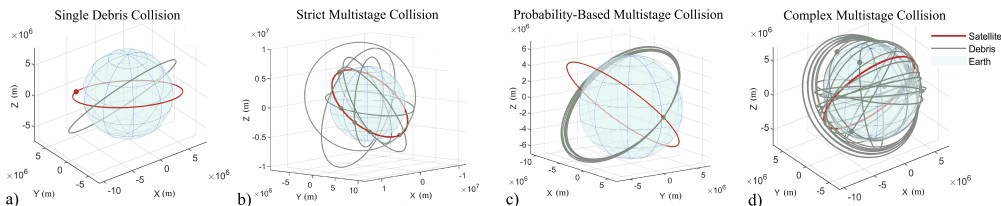

Figure 2: This figure illustrates four collision scenarios of increasing complexity. (a) A single debris collision, with one debris object intersecting the spacecraft's trajectory. (b) A strict multistage collision, where a new debris object is manually added to an existing multi-debris scenario, creating an unavoidable collision. (c) A probability-based multistage collision introduces uncertainty by sampling debris velocities from a normal distribution, forming dense threat clouds. (d) A complex multistage collision integrates sequential strict collisions, probabilistic modeling, and real orbital data from CelesTrak (Kelso, 2025).

### 3.4 REINFORCEMENT LEARNING PROBLEM DEFINITION

The collision avoidance task is formulated as a Markov Decision Process to enable optimization through deep reinforcement learning. The agent observes a time-dependent state vector comprising its own position and velocity, the relative positions, velocities and risk indicators of nearby debris, and then outputs a continuous control vector $u_t$ at the current moment. The state space and action space are as follows:

**1) State Space**: The state space of reinforcement learning can be represented as a collection of the states of spacecraft and space debris, where $\mathbf{r}_t^{\text{agent}}$ and $\mathbf{v}_t^{\text{agent}}$ denote the agent's position and velocity at time $t$, and $\Delta v_{t-1}$ represents the final maneuver vector executed at the previous timestep. For each debris object $i$, $\mathbf{r}_t^i$, $\mathbf{v}_t^i$, $\text{TCA}_t^i$, and $\text{DCA}_t^i$ correspond to its current position, velocity, time of closest approach (TCA), and distance of closest approach (DCA), respectively. Ultimately, the state $s_t$ at time t is:

$$s_t = \{\mathbf{r}_t^{\text{agent}}, \mathbf{v}_t^{\text{agent}}, \Delta v_{t-1}, \{\mathbf{r}_t^i, \mathbf{v}_t^i, \text{TCA}_t^i, \text{DCA}_t^i\}_{i=1}^N\} \qquad (2)$$

**2) Action Space**: Actions are represented as a 3-dimensional continuous control vector $\Delta v_t^{\text{final}} \in \mathbb{R}^3$, constrained by low-thrust limits. The constraint is defined as $\|\Delta v_t^{\text{final}}\| \leq 2.27 \Delta t \times 10^{-4}$ m/s, where $\Delta t$ is the simulation time step.

## 4 THE STAN METHOD

In this section, we introduce the **Spatio-Temporal Attention Network (STAN)** for autonomous satellite collision avoidance in multi-debris environments. STAN is a unified policy that processes

multiple heterogeneous threats to generate both low-thrust maneuvers and task termination decisions. Its modular, attention-based design ensures scalability and physical awareness, consisting of a **multi-head encoding layer**, a physics-aware **spatio-temporal attention fusion layer**, and an **output decoder**. As shown in Figure 1, STAN encodes each spacecraft-debris pair and applies a physics-aware attention mechanism to prioritize high-risk interactions. A lightweight output module is then used to generate a continuous low-thrust maneuver vector, along with a binary signal indicating whether the avoidance task has been completed. Experimental results demonstrate that STAN significantly improves avoidance success rate, reduces fuel consumption, and accelerates policy convergence.

## 4.1 MULTI-HEAD ENCODING LAYER

The input to the network consists of the spacecraft state and a list of debris states. To capture the pairwise interaction between the spacecraft and each debris object, the model concatenates the spacecraft state with each debris state, forming a joint 17-dimensional token vector per pair. These vectors are passed through a shared residual encoder, which consists of two fully connected layers with Tanh activation. This encoder transforms the heterogeneous input sets into a unified, high-dimensional representation space. The output of the encoding layer is a set of $N$ debris state embeddings, denoted as $E \in \mathbb{R}^{N \times D}$, where $N$ is the variable number of debris objects and $D$ is the embedding dimension. This structure ensures permutation invariance and supports variable-length inputs, laying the foundation for scalable multi-object processing in the subsequent attention layer.

## 4.2 SPATIO-TEMPORAL ATTENTION FUSION LAYER

The fusion layer is responsible for aggregating the variable number of debris state embeddings $E$ into a single fixed-size context vector $f \in \mathbb{R}^D$ for decision making. To effectively prioritize high-risk threats, a physics-aware bias is injected, two distinct aggregation strategies are investigated: **Self-Attention Aggregation (SAA)** and **Cross-Attention Aggregation (CAA)**.

### 4.2.1 PHYSICS-AWARE BIAS FORMULATION

Both strategies share a core mechanism: injecting physical risk priors directly into the attention calculations. We define a bias term derived from the Time to Closest Approach ($t_k$) and Distance of Closest Approach ($d_k$) for each debris object $k$. Let $\Phi_k = [t_k, d_k]$ be the physical feature vector for debris $k$. We employ a learnable weight vector $\gamma \in \mathbb{R}^2$ to compute a scalar risk bias $b_k$:

$$b_k = -\gamma \Phi_k. \tag{3}$$

This scalar $b_k$ represents the inherent physical risk of the $k$-th debris object. A higher value indicates higher urgency. This bias is applied to the attention logits before the softmax operation.

### 4.2.2 STRATEGY I: SELF-ATTENTION AGGREGATION (SAA)

The SAA strategy, captures pairwise interactions between all debris objects. The input features $E$ are projected into Query ($Q$), Key ($K$), and Value ($V$) matrices, all in $\mathbb{R}^{N \times D}$. The attention matrix is computed as:

$$A_{\text{self}} = \text{softmax}\left(\frac{QK^T}{\sqrt{D}} + \lambda B_{\text{self}}\right), \tag{4}$$

where $B_{\text{self}} \in \mathbb{R}^{N \times N}$ is the bias matrix broadcasted with $(B_{\text{self}})_{ij} = b_j$. Since softmax normalization is applied row-wise, the addition of this bias will not be canceled. Adding the corresponding bias in each column changes the attention weights of that column's debris relative to all other individuals. This creates a fused feature matrix $F_{\text{self}} = A_{\text{self}}V$. To obtain the single context vector $f$, a pooling operation is required to compress the dimension from $N$ to 1:

**Mean Pooling:** $f = \frac{1}{N}\sum_{i=1}^{N}(F_{\text{self}})_i$. While standard, this may suffer from information dilution in dense environments with sparse threats.

**Max Pooling:** $f = \max_i(F_{\text{self}})_i$. This captures the most salient features but may lose contextual smoothness.

### 4.2.3 STRATEGY II: CROSS-ATTENTION AGGREGATION (CAA)

To overcome the potential information loss associated with pooling operations, we propose the Cross-Attention Aggregation strategy. Instead of computing $N \times N$ interactions, CAA introduces a global, learnable **Maneuver Query** vector $q_{\text{global}} \in \mathbb{R}^{1 \times D}$. This query acts as a task-specific agent searching for critical threats within the debris environment. The debris features $E$ are projected only into Keys ($K$) and Values ($V$). The global attention weights $\alpha_{\text{cross}} \in \mathbb{R}^{1 \times N}$ are computed directly between the global query and debris keys:

$$\alpha_{\text{cross}} = \text{softmax}\left(\frac{q_{\text{global}}K^T}{\sqrt{D}} + \lambda B_{cross}\right), \tag{5}$$

where $B_{cross} = [b_1, ..., b_N] \in \mathbb{R}^{1 \times N}$ is the physics bias vector applied directly to the keys. The final context vector is obtained by a weighted sum:

$$f = \alpha_{\text{cross}}V \tag{6}$$

By bypassing the pooling step, CAA allows the policy to dynamically focus on the most critical object(s) based on the learned query and physical priors. Meanwhile, the addition of N-dimensional bias is also more natural.

### 4.3 OUTPUT DECODER: ACTION AND TERMINATION

The fused context vector $f$ serves as the global summary of the current collision scenario. It is fed into two parallel branches: Action Head and Termination Head.

The **Action Head** predicts the continuous low-thrust maneuver $a$ required to adjust the spacecraft's trajectory. It consists of two fully connected layers with Tanh activations. The output is scaled by the maximum thrust limit mentioned before.

The **Termination Head**, to support adaptive decision-making in multi-stage collision scenarios, processes the fused context through a two-layer MLP with Tanh and sigmoid activations. It outputs a scalar decision variable $p_{\text{done}} \in (0, 1)$, representing the probability that the current avoidance maneuver is complete. This guides the agent to stop thrusting when appropriate, improving fuel efficiency and operational robustness.

Finally, to ensure both physical smoothness and real-time responsiveness, a trajectory-aware smoothing strategy is applied to the final maneuver. Specifically, the previous actions $\Delta v_{t-3}, \Delta v_{t-2}, \Delta v_{t-1}$ are interpolated using a Lagrange polynomial $S(t)$ to estimate the current action trend. The final maneuver vector is then computed as a weighted fusion:

$$\Delta v_t^{\text{final}} = H(0.5 - p_{done})[\alpha S(t) + (1 - \alpha)a], \tag{7}$$

where $H(x) = 1$ if $x > 0$ and $H(x) = 0$ otherwise. $\alpha \in [0, 1]$ is a smoothness coefficient controlling the trade-off between historical continuity and current policy response.

### 4.4 ADVANTAGES OF THE STAN DESIGN

STAN provides several key advantages over conventional collision avoidance policy models. The architecture is **scalable**, supporting a variable number of debris without the need for retraining or reconfiguration. It also has **physical awareness** as it incorporates interpretable proximity and timing metrics into its attention mechanism, allowing it to capture domain-specific decision logic. Finally, it uses a **joint action-termination policy**, integrating maneuver and termination prediction to enable efficient multi-stage reasoning and fuel conservation.

## 5 REINFORCEMENT LEARNING OPTIMIZATION

### 5.1 REWARD FUNCTION DESIGN

To effectively balance safety and efficiency in multistage collision avoidance, we design a composite reward function that integrates multiple mission-critical factors under continuous low-thrust constraints. The immediate reward at time step $t$ is defined as:

$$R_t = r_t + r_f - p_c - p_s - p_f - p_d - p_{\text{dev}} \tag{8}$$

Here, $r_t$ is a small positive survival reward, and $r_f$ is a terminal bonus for successfully avoiding all high-risk debris and reaching the goal. The penalty terms include collision probability ($p_c$), minimum distance softness ($p_s$), fuel consumption ($p_f$), control smoothness ($p_d$), and deviation from the reference orbit ($p_{\text{dev}}$). The collision penalty $p_c$ is based on the cumulative probability $c_i$ of each debris object, with an additional hard penalty applied if $c_{\max} > 0.5$. To mitigate sparse reward issues, a soft penalty $p_s$ is introduced using the predicted minimum distance. Control smoothness is encouraged by penalizing action discontinuity, and $p_{\text{dev}}$ ensures that the agent maintains orbital proximity. This reward design guides the agent to generate fuel-efficient, smooth, and collision-free maneuvers while maintaining orbital stability.

## 5.2 PPO-BASED OPTIMIZATION

The Proximal Policy Optimization (PPO) algorithm, a widely used Actor-Critic method for continuous control tasks, is adopted in this work. The actor network generates maneuver actions under thrust constraints, while the critic estimates the value function to guide the policy update. In our implementation, the critic takes the same encoded and attention-fused structure, and outputs a scalar state-value estimate using a multi-layer perceptron with two hidden layers (128 and 64 units respectively). The advantage estimate is then used to update the actor via clipped surrogate objectives, stabilizing the learning process.

# 6 EXPERIMENTS

## 6.1 EXPERIMENTAL SETTINGS

To ensure diversity, four types of collision scenarios are generated by varying the number, timing, and geometry of the debris, using the aforementioned methods in section 3.3. The simulation is based on two-body orbital dynamics mentioned before and spans a mission window of 1.2 hours, discretized with a fixed timestep of 8.64 seconds. Each avoidance episode lasts up to 500 steps and simulates multiple close approaches with coupled threats. In each training episode, the spacecraft's near-circular orbital velocity vector was randomly initialized tangentially to the geocentric line at a specified orbital altitude, and space debris were randomly initialized along the spacecraft's orbit.

## 6.2 EVALUATION METRICS

To comprehensively evaluate both safety and efficiency, we adopt the following metrics:

**Cumulative Rewards**: The total return accumulated across all time steps, $R = \sum R_t$.

**Collision Probability** (Coll. Prob): For debris $i$, the collision probability $p_i$ is computed in the RSW (Radial, along-Track, Cross-Track) coordinate frame, where $R_i$, $S_i$, $W_i$ are the relative positions, $\sigma_R$, $\sigma_S$, $\sigma_W$ and $\sigma_{SW}$ are standard deviations (Bai et al., 2013):

$$p_i = \exp\left[-\frac{1}{2}\left(\frac{R_i^2}{\sigma_R^2} + \frac{S_i^2 + W_i^2}{\sigma_{SW}^2}\right)\right] \times \left[1 - \exp\left(-\frac{r_A^2}{2\sigma_R \sigma_{SW}}\right)\right] \tag{9}$$

**Fuel Consumption**: Fuel consumption is calculated as the sum of the norms of $\Delta v$:

$$C_{\text{fuel}} = \sum_t \|\Delta v_t^{\text{final}}\| \tag{10}$$

**Orbital Deviation**(Orbit Dev): The L2-norm of the normalized difference between the initial and current osculating elements $\mathbf{x}$, where $\mathbf{n}$ is the normalization vector to balance different units of the orbital elements:

$$C_{\text{dev}} = \left\|\frac{\mathbf{x}_{\text{current}} - \mathbf{x}_{\text{initial}}}{\mathbf{n}}\right\|_2 \tag{11}$$

**Control Rate of Change**(Ctrl RoC.): Measures the smoothness of maneuvers by summing squared differences between consecutive thrust vectors:

$$C_{\text{roc}} = \sum_t \|\Delta v_t^{\text{final}} - \Delta v_{t-1}^{\text{final}}\|^2 \tag{12}$$

**Training Time**: Total time required to train the policy until convergence.

**Inference Time**: Total runtime for a fixed number of decision steps.

## 6.3 ABLATION STUDY

To rigorously evaluate the contribution of each component in STAN and validate the proposed aggregation strategy, we conducted comprehensive ablation studies in the probability-based 10-debris multistage collision scenario. To ensure statistical reliability, all experiments were repeated using 5 distinct random seeds (42, 422, 4222, 42222, 422222). In the reported results (Figure 3), solid lines represent the mean reward across seeds, while the shaded regions indicate the standard deviation.

First, we compared the full STAN model against three variants: 1) **w/o Termination**: removes the termination head, forcing the agent to act continuously; 2) **w/o ST-Attention**: cancels the physical bias, and replaces ST-Attention with standard scaled dot-product attention; 3) **Vanilla Transformer**: a baseline stripped of both the termination head and the ST-Attention mechanism. As shown in Table 1 and part i of Figure 3, both the **w/o Termination** variant (blue) and the **Vanilla Transformer** (yellow) achieve higher initial and final mean rewards than the full STAN model and its other variants. However, this high raw reward may be misleading, as these policies lack the ability to stop thrusting, leading to significant fuel waste (be further quantified in Figure 5). The full STAN model (purple) learns to balance collision avoidance with the termination decision, optimizing operational efficiency. Comparing the models with ST-Attention to those without, the introduction of the physics-aware bias mechanism does not lead to a significant difference in the final converged reward. Nevertheless, during the intermediate learning phase, the inclusion of ST-Attention demonstrably accelerates the model's learning speed and stabilizes the reward growth trajectory.

Next, for aggregation, we compared SAA and CAA strategies. In SAA, we compare average pooling and maximum pooling. Part ii of Figure 3 shows that the Cross-Attention approach exhibits slightly faster initial learning and a marginally higher final reward. Interestingly, Max Pooling learns faster than Mean Pooling, but both ultimately reach a similar performance level in this scenario. Three methods do not show significant disparities in their final converged rewards. We hypothesize this is due to the nature of the simulated environment: the debris cluster generated based on a single probability distribution often results in objects that are relatively close to each other, with hazard levels that do not vary drastically, which limits the efficacy gains of aggregation strategies.

Finally, for the interpretation of Learned Attention Components. Figure 4 visualizes the two components of the cross-attention score of CAA strategy: the learned interaction term $q_{\text{global}} K^\top / \sqrt{D}$ and the physics-based bias term $\lambda B_{\text{cross}}$, for 10-debris scenario mentioned before. The physics-aware term exhibits markedly larger magnitude than the learned logits, consistent with the trained parameters $\lambda = 0.0993$ and $\gamma = [0.9891, , 0.9970]$. This indicates that the final attention weights rely strongly on the injected physical priors. Both $t_k$ and $d_k$ contribute to the bias, though $d_k$ plays a slightly more influential role. Notably, debris indices 2, 5, and 8 receive consistently high values in both components, showing that the learned query and the physics prior agree in identifying these objects as critical threats.

Table 1: Performance comparison in evaluation of 10-debris probability-based multistage collisions

| Type | Rewards | Coll. Prob | Fuel Cons. | Orbit Dev. | Ctrl RoC. | Train (s) | Infer (s) |
|------|---------|-----------|-----------|-----------|-----------|-----------|-----------|
| STAN | 504.19 | 7.25E-20 | 0.355 | 0.336 | 8.20E-05 | 9616 | 3.163 |
| w/o Term. | 472.78 | 2.00E-61 | 0.651 | 1.055 | 5.10E-05 | 9344 | 3.109 |
| w/o ST-Att. | 467.678 | 9.55E-23 | 0.360 | 1.055 | 7.1E-05 | 9080 | 3.082 |
| Transformer | 419.75 | 1.12E-02 | 0.618 | 0.336 | 3.30E-05 | 9778 | 3.137 |
| FC | 394.56 | 2.22E-132 | 0.759 | 1.055 | 2.10E-05 | 11809 | 3.208 |
| CNN | 56.52 | 3.76E-16 | 0.245 | 1.055 | 6.90E-05 | 9731 | 3.121 |
| LSTM | -102.17 | 6.25E-02 | 0.146 | 1.055 | 6.20E-05 | 8245 | 3.131 |

## 6.4 BASELINE COMPARISON

To further validate the effectiveness of the proposed **STAN** architecture, we compared it against three baseline models: an **LSTM**-based network, a **CNN**-based network, and a fully connected (**FC**) net-

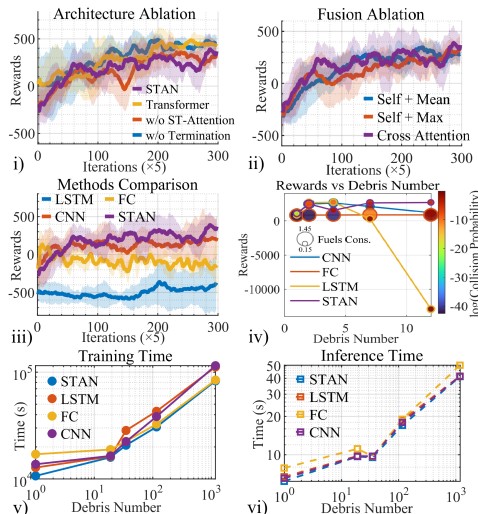

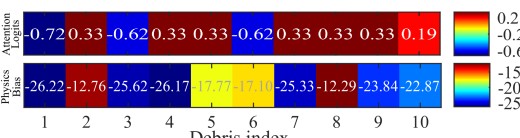

Figure 4: Comparison of learned attention logits and physics-based bias values across 10 debris.

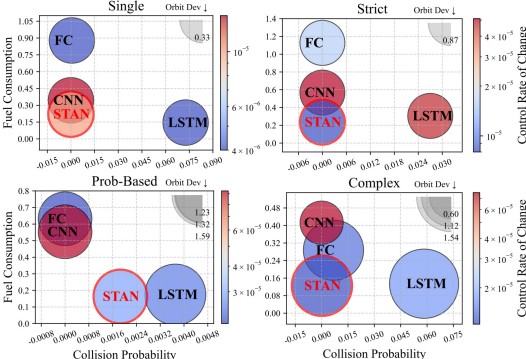

Figure 3: (i) Training curves for ablation. (ii) Training curves for different aggregation strategies. (iii) Training curves for baseline comparison. (iv) Generalization performance of models trained on 3-debris scenarios, evaluated on 1-19 debris. (v) Training time under different debris number. (vi) Inference time under different debris number.

Figure 5: Bubble chart for baseline comparison. X-axis denotes collision probability, Y-axis denotes fuel consumption. Bubble size reflects orbital deviation, and color represents the control rate of change(bluer means smoother).

work. The evaluation was conducted across the four representative collision scenarios described above: Single, Strict Multistage, Probability-Based, and Complex. To ensure statistical reliability, all experiments were repeated using 5 distinct random seeds (42, 422, 4222, 42222, 422222). In the reported results, solid lines represent the mean reward across seeds, while the shaded regions indicate the standard deviation. Part iii of Figure 3 shows the **learning curves** obtained in the probability-based 10-debris multistage collision scenario. STAN consistently outperformed the baselines in terms of cumulative rewards and convergence speed. Ultimately, STAN achieved a 22% improvement in cumulative rewards over CNN, a 142% improvement over FC and a 746% improvement over LSTM.

The **generalization performance** was evaluated by training all models on strict 3-debris multistage scenarios and testing them across 1–19 debris. As shown in part iv of Figure 3 , STAN, CNN, and FC successfully avoided collisions, while LSTM failed in stages 12–19. STAN achieved rewards consistently above 1000, collision probabilities below 0.08, and fuel consumption around 0.2—nearly 85% lower than FC.

For both **training and inference time**, all networks were trained for 1,500 steps using the same timestep settings across identical scenarios on a system equipped with an Intel Core i7 processor and an NVIDIA RTX 3060 graphics card. As shown in Table 1, STAN also ranked among the best-performing methods in this aspect. In part v and vi of Figure 3, the x-axis represents the number of debris objects, while the y-axes denote training time and inference time. The results indicate that STAN consistently achieved faster execution compared to the others.

The **test results** for four scenarios are shown in Figure 5, where the x-axis indicates collision probability, y-axis shows total fuel consumption, bubble size reflects orbital deviation, and color represents control rate of change (bluer means smoother). STAN consistently occupies the optimal lower-left region, maintaining collision probability less than 0.002, fuel consumption less than 0.3, and deviation less than 0.32. LSTM performs worst with bubbles on the right. CNN suffers from poor control smoothness (redder bubbles) and a control rate up to $6 \times 10^{-5}$. FC surpasses CNN in success rate but shows 436% higher fuel consumption than STAN (STAN saves 81% fuel), reflected in its higher-positioned bubbles. These findings confirm that STAN achieves a better balance between safety and efficiency and is especially advantageous in complex multistage collision scenarios.

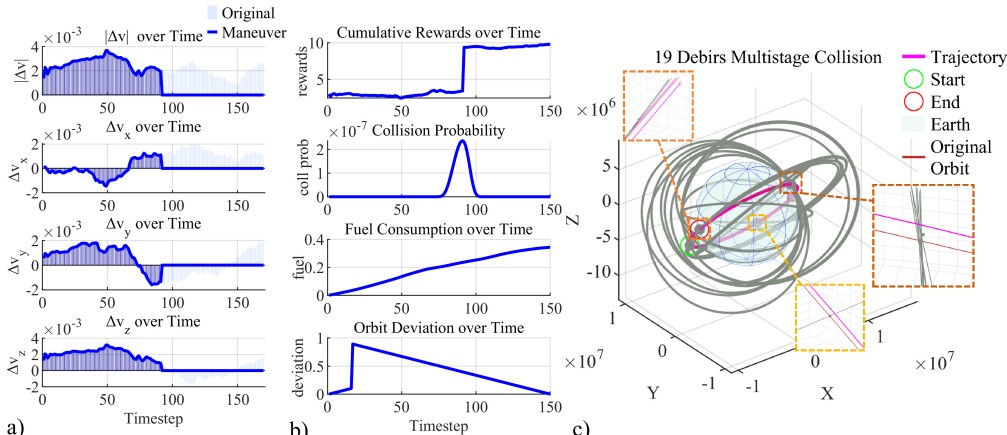

Figure 6: Results of 19-debris strict multistage avoidance: a) Maneuver vectors, from top to bottom are maneuvers' modulus values and the components in each direction b) Variation of rewards, collision probability, fuel consumption and orbit deviation over time c) Figure of the avoidance trajectory for 19-debris strict multistage collisions, STAN avoided collisions at all intersection points

### 6.5 CASE STUDY: STRICT 19-DEBRIS MULTISTAGE COLLISIONS

To further illustrate the effectiveness of STAN, we present a case study of a strict multistage collision scenario involving 19 debris. As shown in part a of Figure 6, the STAN controller generates smooth and continuous control actions, with deep blue segments indicating executed maneuvers and light blue representing skipped actions based on termination prediction. This leads to improved control smoothness and approximately 45% fuel savings. Part b shows the evolution of cumulative reward, collision probability, fuel consumption, and orbital deviation during the task, demonstrating stable learning and reliable avoidance. Part c visualizes the resulting avoidance trajectory, where the spacecraft successfully avoids all debris threats. This case confirms that STAN can handle highly complex collision scenarios with efficient and safe maneuvering.

## 7 CONCLUSION

In this work, autonomous collision avoidance for spacecraft in high-risk, multi-debris environments under **low-thrust constraints** was investigated. A spatio-temporal attention network (STAN) was introduced, leveraging a physics-aware multi-head attention mechanism to extract and integrate dynamic spatiotemporal features. STAN enables accurate termination prediction and efficient control, demonstrating strong adaptability to variable-length multistage collision scenarios.

Ablation results showed that ST-Attention improved cumulative rewards by 184% compared to traditional attention. Through extensive experiments across representative baseline architectures (fully connected networks, CNNs, LSTMs), STAN consistently achieved superior performance. It maintained collision probabilities below 0.002, reduced control rate variations, and achieved up to 81% fuel savings compared to baseline methods, and is capable of handling complex multistage collision scenarios with up to 116 debris. Case studies and generalization tests further confirmed its robustness. The key contributions of this study are a proposed physics-inspired **spatio-temporal attention** mechanism (ST-Attention), the development of the **STAN architecture** which is capable of processing variable-length input and autonomously terminating maneuver outputs, and the construction of challenging test environments to evaluate STAN across multiple baselines.

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

# A APPENDIX

## A.1 EARLY WARNING METHODS AND EFFECT FOR DEBRIS COLLISIONS

To efficiently manage the computational load associated with a large number of space objects, we implement a robust two-stage early warning screening process. This hierarchical approach systematically filters potential threats to identify high-risk conjunctions for the primary spacecraft, ensuring that computational resources are focused only on relevant objects. The early warning methods mentioned above consist of two-stage collision risk screening: static screening based on orbital altitude, and dynamic screening based on crossing time.

In the **static screening**, all space object orbits are first propagated using the two-body model. Let $h_{p,\text{main}}$ and $h_{a,\text{main}}$ denote the primary spacecraft's perigee and apogee. A debris object is filtered out if it does not intersect the spacecraft's altitude range, beyond a margin d: $h_{p,\text{main}} - h_a > d$ or $h_p - h_{a,\text{main}} > d$. This step efficiently removes objects with no spatial overlap.

Next, **dynamic screening** is performed based on the time of passage through orbital intersection points. By computing orbital plane normals and their cross product, the line of intersection is determined. The orbital period $T_s$ is calculated as:

$$T_s = 2\pi \sqrt{\frac{a^3}{\mu}} \tag{13}$$

Given the first crossing time $t_1$, subsequent passage times are estimated as $t_i = t_1 + (i-1)T_s$. Any object for which the passage time difference satisfies $|t_j - t_k| > t_c$ (where $t_c$ is a temporal threshold) is eliminated. Only debris with both spatial and temporal proximity are retained for risk assessment. This two-stage method substantially reduces the number of candidate threats, improving efficiency for downstream learning and planning. The effectiveness of this two-stage screening approach is illustrated in Figure 7, which demonstrates the reduction of potential collision targets through successive filtering.

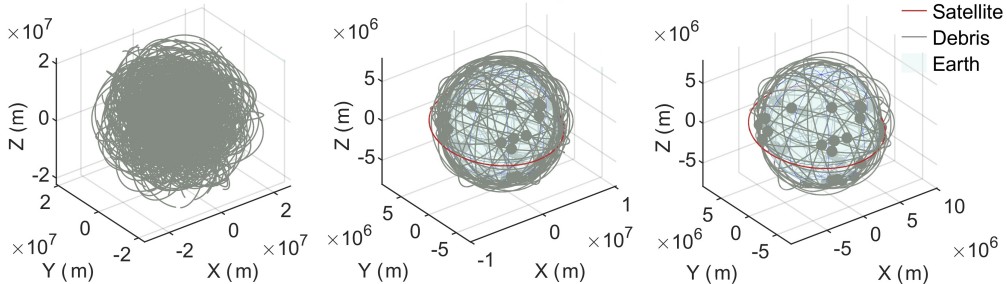

Figure 7: Screening effect comparison

## A.2 SCENE CONSTRUCTION METHODS AND SCENE PARAMETERS

To thoroughly evaluate the robustness and adaptability of our proposed model, we designed four distinct and challenging collision avoidance scenarios. The construction of these scenarios is crucial for training and testing the agent's decision-making capabilities under various conditions. We developed two primary methods for generating these environments: a deterministic approach for creating strictly staged encounters and a probabilistic approach for simulating more chaotic, realistic debris fields.

Algorithm 1 details the procedure for constructing the Strict Multistage Collision Scenario. This method iteratively generates debris objects at fixed time intervals ($\Delta t_c$) along the spacecraft's planned trajectory. It creates a highly structured problem where the agent must sequentially resolve a series of predictable, high-risk encounters, testing its ability to plan long-term avoidance strategies.

In contrast, Algorithm 2 describes the method for generating the Probability-Based Multistage Collision Scenario. This approach samples debris velocities from a normal distribution, creating a cluster

---

**Algorithm 1** Strict Multistage Collision Scenario Construction

---

**Input**: Initial orbit of the primary spacecraft $X$, first collision time $t_p$, collision interval $\Delta t_c$, total number of collisions $n$
**Output**: Training scenario with $n$ staged collision events

    Initialize counter $i \leftarrow 0$, current time $t \leftarrow t_p$
    Set the avoidance trajectory to the original orbit of the primary spacecraft
    **while** $i < n$ **do**
      Generate a new debris object at position corresponding to time $t$ along the current avoidance trajectory
      **if** the agent fails to avoid collisions with all existing debris **then**
        Continue training the agent until successful avoidance is achieved
      **end if**
      Update the avoidance trajectory using the agent's latest control strategy
      $i \leftarrow i + 1; \quad t \leftarrow t + \Delta t_c$
    **end while**
    **return** the constructed $n$-stage collision avoidance scenario

---

of debris with varied trajectories around a central collision point. This method simulates a more unpredictable environment, akin to the fragmentation of a larger object, and tests the agent's ability to handle multiple simultaneous threats.

---

**Algorithm 2** Probability-Based Multistage Collision Scenario Construction

---

**Input**: Initial orbit of the primary spacecraft, collision time $t_p$, number of debris objects $n$, velocity distribution $N(\mu, \sigma)$
**Output**: A training scenario with $n$ debris objects generated from a probabilistic velocity model

1: Sample $n$ velocity vectors $v_i \sim N(\mu, \sigma)$ for $i = 1, 2, \ldots, n$
2: **for** each $v_i$ **do**
3:     Generate a debris object with velocity $v_i$ at the position of the primary spacecraft at time $t_p$
4: **end for**
5: **return** the constructed probability-based multi-debris collision scenario

---

Table 2 summarizes the key parameters defining the four scenarios used in our experiments, including the number of debris objects and the range of orbital altitudes. These scenarios—Single, Strict Multistage, Probability-Based, and Complex—provide a comprehensive benchmark for evaluating model performance, ranging from simple one-on-one encounters to highly congested orbital environments.

Table 2: Scenario settings for collision avoidance evaluation

| Scenario Type | No. of Debris | Orbital Altitude(km) |
|---|---|---|
| Single | 1 | 519 |
| Strict Multistage | 2-19 | 400-800 |
| Probability-Based | 3-40 | 350-1100 |
| Complex | 1178 | 300-2000 |

A.3    REPRODUCIBILITY STATEMENT

The training of our deep reinforcement learning agent was conducted using the Proximal Policy Optimization (PPO) algorithm. The stability and performance of the training process are highly dependent on the choice of hyperparameters. Table 3 provides a comprehensive list of the specific hyperparameters and environmental settings used in our experiments. These values were carefully selected through empirical tuning to ensure efficient convergence and optimal policy learning. Key

parameters include the learning rate, discount factor ($\gamma$), and the clipping parameter ($\epsilon$), which are critical for balancing exploration and exploitation while maintaining stable policy updates.

Table 3: Hyperparameters and Environment Settings for PPO Training

| Parameter | Value |
|---|---|
| Propagation Step $\Delta t$(Day) | 0.0001 |
| Action Std (Initial) | $0.003 \times dV_{limit}$ |
| Learning Rate $\alpha$ | 0.0003 |
| Optimizer | AdamW |
| Optimizer's Betas | (0.9, 0.999) |
| K Epochs per Update | 3 |
| Clipping Parameter $\epsilon$ | 0.2 |
| Discount Factor $\gamma$ | 0.99 |
| Update Interval $\Delta t_{\text{update}}$ | 1000 |
| Batch Size | 300 |
| Learning Rate Schedule | Linear decay, $\max(0.1, 1 - \frac{\text{step}}{10000})$ |
| Minimum Learning Rate Ratio | 0.1 |
| Action Std Decay Threshold | $0.001 \times dV_{\text{limit}}$ |
| Action Std Decay Rate | 0.99 |

## A.4 Additional Experimental Results

In this section, we present additional experimental results regarding the training and evaluation processes of STAN and the compared baseline methods across the various scenarios outlined earlier. These experimental results consistently demonstrate that STAN exhibits better learning performance: it not only achieves higher cumulative rewards but also generates substantially smoother low-thrust control signals.

### A.4.1 Training Process and Results

In our experiments, we trained STAN and other baseline methods across the four types of multistage collision scenarios detailed in the main text. The main text already includes the training curve for the 10-debris probability-based multistage collision scenario. Here, we present the results for the remaining three scenarios, with the experimental outcomes illustrated in Figure 8.

As shown in part a of Figure 8, in the single-debris collision scenario, among all compared methods (LSTM, CNN, FC, and variants without ST-Attention or Termination), STAN consistently achieves the highest and most stable cumulative rewards. While other baselines (e.g., LSTM, CNN, FC) show significant fluctuations or lower reward ceilings, STAN's reward curve rapidly converges to a high value and maintains stability throughout training. This demonstrates STAN's superior ability to learn optimal control policies in relatively simple, single-debris collision contexts.

In part b of Figure 8, which depicts the strict multistage collision scenario with 19 debris, the sequential avoidance difficulty is substantially heightened by the scenario's defining constraint: each successful collision avoidance forces the next collision event to occur directly on the resulting evasion trajectory. Notably, unlike the single-debris scenario or the probabilistically generated 10-debris scenario in the main text (which exhibit significant reward fluctuations due to random environment regeneration), this fixed scenario structure produces training curves with smaller step-to-step variations. However, distinctive sudden drops into low reward regions—followed by gradual recovery phases—are observed. This pattern likely stems from the strict sequential collision rules: when the agent successfully evades one debris, the pre-programmed emergence of a new collision on its adjusted trajectory frequently leads to unexpected secondary collisions, causing abrupt reward declines. As training progresses, the agent gradually learns to anticipate these chained events, enabling recovery of performance over subsequent iterations. Most baseline models (LSTM, CNN, and FC) fail to stabilize under this regime, often converging to suboptimal or even negative rewards. Removing key components from STAN (i.e., without ST-Attention or without Termination) also results in unstable or degraded performance—particularly struggling to recover from those sudden drops—highlighting the critical role of these mechanisms in handling sequential dependencies. In

contrast, the complete STAN model exhibits more robust recovery dynamics and ultimately achieves consistently high rewards, demonstrating its superior capability to navigate the chained collision challenges inherent to this scenario.

In part c of Figure 8, the complex multistage collision scenario with 116 debris—constructed using real-world debris distributions—poses the most challenging environment. Most baselines suffer from severe instability or collapse (e.g., CNN and FC), and even recurrent models such as LSTM cannot maintain reliable performance. Ablated STAN variants show noticeable drops compared with the full model, further confirming the necessity of spatio-temporal attention and termination design. The full STAN achieves fast convergence to near-optimal rewards and maintains stability across the entire training process, underscoring its scalability and effectiveness in highly realistic large-scale collision-avoidance problems.

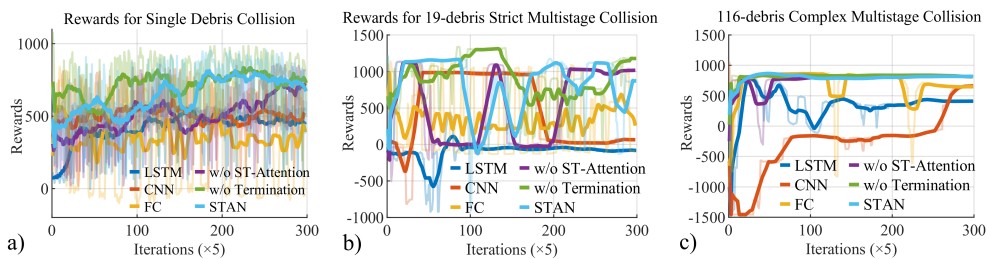

Figure 8: Training Curve Comparison

### A.4.2 EVALUATION RESULTS

This section presents a detailed quantitative analysis of the performance of our proposed STAN model compared to three baseline architectures: a Fully Connected (FC) network, a Convolutional Neural Network (CNN), and a Long Short-Term Memory (LSTM) network. Figure 9 presents a comparative analysis of maneuver sequences generated by six models during the testing phase in the 19-debris strict multistage collision scenario. Each subplot displays four components from top to bottom: the magnitude of the maneuver ($|\Delta v|$), and the maneuver components in the $x$, $y$, and $z$ directions, respectively.

For the full STAN model (subplot a), the control signals exhibit smooth and continuous profiles across all dimensions. Notably, after successfully evading debris, STAN can autonomously terminate the maneuver output, which is crucial for minimizing fuel consumption in electric propulsion satellites relying on continuous low-thrust control. In contrast, the ablation variants of STAN (subplots b and c, without Termination and without ST-Attention, respectively) show slightly less optimal performance. While their maneuver patterns are relatively smoother than the baseline methods, they lack the ability to precisely cut off thrusts post - avoidance, leading to more unnecessary fuel usage compared to the full STAN model.

Turning to the baseline methods (subplots d–f), significant drawbacks emerge. The CNN (subplot d) and LSTM (subplot f) models produce highly fluctuating maneuver signals, indicating unstable and inefficient control strategies. The FC model (subplot e), although generating maneuvers with less abrupt jumps, consistently outputs constant - magnitude thrusts without adaptive variations. This static control behavior raises serious concerns about whether the FC model has truly learned effective collision - avoidance strategies, as it fails to adjust thrusts dynamically according to the evolving collision risks. Overall, these results highlight STAN's superiority in enabling smooth, fuel - efficient, and adaptive low - thrust control for debris avoidance tasks.

Table 4 summarizes the key performance metrics across the four distinct collision scenarios. The metrics include training and inference times, final reward, post-maneuver collision probability, fuel consumption (normalized), final orbital deviation (km), and the rate of change of control actions. The corresponding data are summarized in the bubble chart in the main text, providing a comprehensive empirical basis for evaluating the effectiveness of each model with respect to safety, efficiency, and orbit preservation.

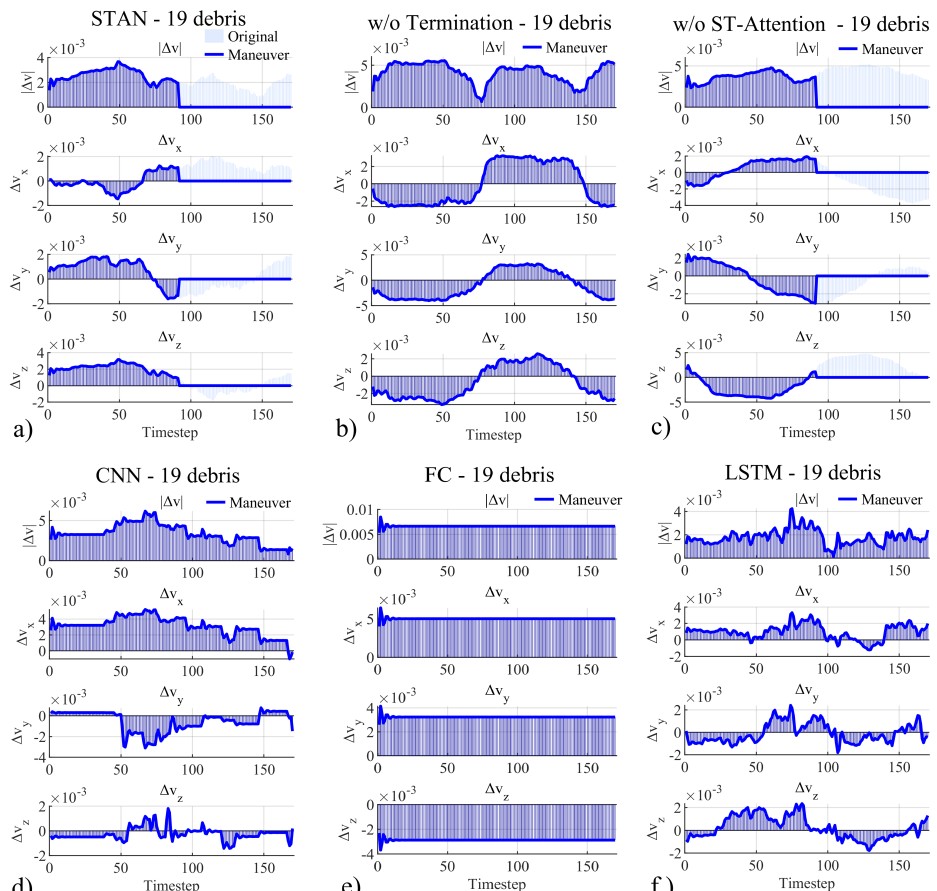

Figure 9: Maneuver Sequence Comparison under 19-Debris Strict Multistage Collision

Table 4: Performance comparison under different scenarios

| Type | Scenario | Train(s) | Inference(s) | Rewards | Coll. Prob | Fuel Cons. | Orbit Dev. | Ctrl RoC. |
|------|----------|----------|--------------|---------|------------|------------|------------|-----------|
| STAN | Single | 1.1e4 | 6.2 | 1.9e3 | 5.8e-16 | 0.22 | 0.33 | 1.0e-5 |
| STAN | Strict | 1.6e4 | 9.7 | 1.0e3 | 2.4e-7 | 0.24 | 0.87 | 8.0e-6 |
| STAN | Probability | 1.4e4 | 7.5 | 7.8e2 | 1.9e-3 | 0.16 | 1.2 | 2.9e-5 |
| STAN | Complex | 8.3e4 | 41 | 7.2e2 | 1.7e-4 | 0.13 | 1.2 | 1.2e-5 |
| FC | Single | 1.7e4 | 7.9 | 1.7e3 | 6.2e-4 | 0.88 | 0.33 | 4.0e-6 |
| FC | Strict | 1.9e4 | 11 | 1.1e3 | 2.8e-21 | 1.1 | 0.87 | 1.6e-5 |
| FC | Probability | 1.4e4 | 7.7 | 8.3e2 | 1.0e-38 | 0.63 | 1.2 | 2.1e-5 |
| FC | Complex | 8.4e4 | 51 | 6.5e2 | 6.7e-3 | 0.29 | 1.2 | 1.4e-5 |
| CNN | Single | 1.4e4 | 6.5 | 1.2e3 | 2.2e-14 | 0.34 | 0.33 | 1.4e-5 |
| CNN | Strict | 1.6e4 | 9.7 | 8.9e2 | 2.6e-12 | 0.57 | 0.87 | 5.1e-5 |
| CNN | Probability | 1.4e4 | 7.8 | 7.6e2 | 3.4e-7 | 0.55 | 1.2 | 9.3e-5 |
| CNN | Complex | 1.1e5 | 41 | 6.9e2 | 1.3e-8 | 0.41 | 0.60 | 7.6e-5 |
| LSTM | Single | 1.3e4 | 6.7 | 1.2e3 | 7.3e-2 | 0.15 | 0.33 | 4.0e-6 |
| LSTM | Strict | 1.6e4 | 9.8 | 9.7e2 | 2.7e-2 | 0.31 | 0.87 | 4.8e-5 |
| LSTM | Probability | 2.1e4 | 7.7 | 8.3e2 | 3.7e-3 | 0.17 | 1.6 | 2.5e-5 |
| LSTM | Complex | 1.1e5 | 42 | 1.6e2 | 5.8e-2 | 0.13 | 1.5 | 1.7e-5 |

### A.5 USE OF LARGE LANGUAGE MODELS IN MANUSCRIPT PREPARATION

In the preparation of this manuscript, large language models (LLMs), specifically *Gemini 2.5 Flash* and *ChatGPT 5*, were utilized as assistive tools to enhance the quality of the writing and support the research process. The application of these models was strictly limited to the following two areas:

- **Writing Assistance:** The LLMs were employed to aid in polishing the manuscript's prose. This included tasks such as improving grammar, rephrasing sentences for enhanced clarity and readability, and ensuring stylistic consistency. The core ideas, arguments, and scientific contributions presented herein remain entirely the work of the authors.

- **Information Retrieval and Discovery:** The LLMs were used as preliminary tools to aid in literature discovery. Their role was to help formulate effective search queries and to identify potential avenues for exploring related work. It is important to note that every source and piece of information suggested by the LLMs was subsequently verified for accuracy and relevance using traditional academic databases and critically evaluated by the authors.

The authors take full responsibility for the intellectual content of this paper, including the accuracy of all data, the validity of the arguments, and the proper citation of all sources. The LLMs served solely as productivity tools and are not credited as authors.

### A.6 ETHICS STATEMENT

The primary objective of this research is to address a critical challenge for the sustainable use of space: the mitigation of collision risks from orbital debris. Our work is intended to enhance the safety of space operations, protect valuable scientific and commercial assets, and contribute to the long-term viability of the near-Earth orbital environment for all.

We acknowledge that the autonomous maneuvering technology developed in this work, like many advanced control systems, could have the potential for dual-use. To address this, we affirm that our research is exclusively focused on the peaceful and defensive application of safeguarding spacecraft from unintentional collisions. We advocate for the responsible development and deployment of such technologies under strict ethical guidelines and human oversight.

Furthermore, we recognize the importance of AI safety and reliability. The AI model was rigorously trained and evaluated across a wide range of simulated scenarios designed to test its robustness and prevent unintended negative consequences. We recommend that any real-world implementation of this system operate within a "human-in-the-loop" framework, where human operators retain the final authority for critical decisions. Our commitment is to foster transparency and to ensure that our contributions positively support a safe and sustainable space for global benefit.

