# OpenReview forum: "STAN: A Spatio-Temporal Attention Network for Space Debris Multistage Collision Avoidance"
_ICLR.cc/2026/Conference — Submitted to ICLR 2026_

### Official Review · Reviewer_gUnw · 2025-10-22

**Soundness:** 2
**Presentation:** 2
**Contribution:** 2
**Rating:** 4
**Confidence:** 3

**Summary:**

The paper introduces **STAN**, a reinforcement‑learning policy for satellite–debris collision avoidance under continuous low‑thrust. The key architectural idea is a Spatio‑Temporal Attention (ST‑Attention) layer that injects a physics‑motivated bias—computed from each object’s distance of closest approach (DCA) and time to closest approach (TCA)—into scaled dot‑product attention over a variable‑sized set of debris. STAN outputs both a thrust vector and a termination signal; training uses PPO across four scenario families (single, strict multistage, probability‑based multistage, complex). The manuscript claims large improvements in reward, fuel, and collision probability versus FC/CNN/LSTM baselines, with ablations for attention and termination.

**Strengths:**

I think the problem choice is timely and practically important: multistage conjunction handling with **continuous** low-thrust and variable-length inputs is closer to real electric-propulsion operations than impulsive-burn abstractions. The physics-aware inductive bias—folding DCA/TCA into attention—offers an intuitive way to prioritize likely threats without heavy modeling; the mechanism is clearly written with $b_i=-\sum_j\gamma_j\Phi_{ij}$ and $W=\mathrm{softmax}(QK^\top/\sqrt{D}+\lambda B)$. Conceptually, the **termination head** is a good interface for fuel economy in continuous-thrust settings. I also appreciated the scenario taxonomy and visuals, and that some PPO hyperparameters and environment details are spelled out. In terms of significance, a robust policy for variable-$N$ debris could matter for on-board autonomy. And in terms of clarity, the high-level pipeline and figures communicate the intent well, even if some equations/indices need tightening.

**Weaknesses:**

**1) Termination gating appears inverted and non-differentiable.**
Equation (7) computes the final action as
$ \Delta v_t^{\text{final}} = H(p_{\text{done}}-0.5)\,[\alpha S(t)+(1-\alpha)a] $,
which *enables* thrust when the model believes the task is “done,” contradicting the prose (“stop thrusting when appropriate”) and discarding the continuous probability via a hard Heaviside. I strongly recommend flipping the logic (e.g., multiply by $1-p_{\text{done}}$ or use a smooth sigmoid gate) and **re-running all experiments**.

**2) ST-Attention bias broadcasting is ambiguous and may be ineffective.**
You define a per-debris bias $b_i$ from $\Phi_i=(d_i,t_i)$ and then state “each row is identical ($B_{ij}=b_j$)” before adding $\lambda B$ to the logits. If $B_{i*}$ is row-constant, adding the same constant to a row cancels in the row-wise softmax; if it’s column-constant (per-key), the bias is global and not pairwise. Please pin down the tensor shapes and show an ablation that the bias actually changes attention maps and outcomes.

**3) Internal inconsistencies weaken trust.**
The manuscript alternates between a collision-probability threshold of $0.02$ vs $0.002$; the ablation-section narrative claims rewards “around $750$” in the 10-debris probabilistic setting while the table lists $\approx 276$; scenario size is “up to $116$” debris in one place and $1178$ elsewhere. Some tables also show FC outperforming STAN on reward (and reporting implausibly tiny probabilities) despite the text claiming consistent dominance. Please take a look at these items.

**4) Collision-probability ($P_c$) modeling is under-specified and numerically suspect.**
The analytic form lacks the covariance definitions $\sigma_R,\sigma_S,\sigma_W,\sigma_{SW}$ and how they are estimated/propagated. Reported values span $10^{-34}$–$10^{-1}$, which looks implausible without a careful uncertainty model and numerically stable evaluation. Safety claims based on $P_c$ are hard to interpret or reproduce.

**5) Baselines do not represent current best set/attention policies (and may be unfair).**
Given STAN’s set-attention core, comparisons against FC/CNN/LSTM are not enough. Please include capacity-matched **vanilla Transformer/Set Transformer** and consider a simple **model-based** low-thrust planner, with clearly documented padding/masking for permutation robustness.

**6) Physics realism gap.**
Everything uses a two-body model, yet “complex” scenarios reference real-world (e.g., CelesTrak) data. Over a 1.2-hour window some regimes may be fine, but ignoring J2/drag (especially in LEO) and omitting the TLE→state pipeline (e.g., SGP4) risks mischaracterizing DCA/TCA and $P_c$.

**7) Methodology and statistics need tightening.**
Results appear single-seed; no mean ± std or clear train/test splits. The “strict multistage” scenario is policy-dependent (training continues until success, then the next collision is injected on the updated trajectory), which risks leakage between training and evaluation. Runtime scaling claims should be backed by matched FLOPs/wall-clock under identical widths/batching.

**8) Reproducibility and clarity gaps.**
Key actor sizes, number of heads/embedding $D$, normalization/dropout, and values/schedules for $\lambda,\gamma,\alpha$ are missing. Orbit deviation is defined on normalized elements but table captions/values sometimes read like kilometers.

**Questions:**

Every item raised in the Weaknesses section can be viewed as a question for the authors.
I may well be mistaken on several of these points, and I would sincerely appreciate clarification or correction wherever appropriate.
If the authors can address or resolve even part of these concerns—whether by showing that I misunderstood something or by providing additional detail—it would be very helpful.

---

> ### Author Response · Authors · 2025-11-17
> **Response to Reviewer gUnw**
>
> ## 1. Termination gating and differentiability
>
> We sincerely thank the reviewer for raising this important point. The equation in the paper indeed contains a sign mistake, and we appreciate the reviewer’s careful reading. After double-checking our implementation, we confirm that the code follows the correct logic—the agent stops thrusting when $p_{\text{done}}$ is high, consistent with the intended behavior. We have verified that the implementation is correct and the gating logic behaves as intended.
>
> Regarding the use of a hard Heaviside gate: we originally experimented with a smooth formulation such as scaling the thrust by $(1 - p_{\text{done}})$, but this consistently **reduced avoidance performance**, likely because partial thrust during an unfinished avoidance maneuver weakened the agent’s ability to respond decisively. This led us to adopt a simpler binary “ongoing vs. done” termination gate, which performed better in practice.
>
> ## 2. Attention bias broadcasting
>
> Thanks very much for pointing out the ambiguity regarding the broadcasting of the ST-Attention bias. In our implementation, each debris object i produces a scalar bias $b_i$ from its physical features $\phi_i$. These biases are arranged in the matrix
> $$
>  B =
>  \begin{bmatrix}
>  b_1 & b_2 & \cdots & b_N \\\\
>  b_1 & b_2 & \cdots & b_N \\\\
>  \vdots & \vdots & & \vdots \\\\
>  b_1 & b_2 & \cdots & b_N
>  \end{bmatrix}
> $$
> which is **column-constant (per-key)** rather than row-constant. Thus, each column j corresponds to the physical importance of debris j as a *key*, and the bias affects all queries consistently. Because the softmax normalization is applied row-wise, adding a constant *per column* does *not* cancel out. In contrast, a row-constant matrix would indeed vanish under softmax; our formulation avoids that issue.
>
> Ideally, a fully pairwise bias $B_{ij}$ would model interaction-specific relationships between debris i and debris j. However, in our setting each token represents only the interaction between debris i and the spacecraft, and the available physical features are limited to TCA/DCA with respect to the spacecraft. Constructing a true pairwise matrix $ \phi_{ij} $ would require computing debris-to-debris encounter metrics (pairwise TCA/DCA for all (i,j)), which is computationally expensive and becomes prohibitive in high-debris environments. For the spacecraft–debris collision-avoidance task considered in this work, debris-debris interaction does not affect spacecraft risk and would not change the action policy.
>
> We fully agree that in multi-agent spacecraft avoidance or scenarios where debris–debris interactions matter, a pairwise physical bias would be valuable and necessary. We have added clarifications in the revised manuscript and will include an ablation demonstrating that the per-key bias does change the attention distribution and improves performance.
>
> ## 3. Internal inconsistencies
>
> Thank you for pointing out the inconsistencies in thresholds, reported rewards, and scenario sizes.    We will carefully correct all numerical values, ensure internal consistency across text and tables, and revise the discussion where needed.
>
> ## 4. Collision-probability modeling
>
> We agree that the current description of the covariance terms and propagation is insufficient.    In the revision, we will provide clearer definitions, improve the numerical treatment of the uncertainty model, and add validations to ensure that the reported probability ranges are physically plausible.
>
> ## 5. Baselines
>
> Thank you for the suggestion.    We will add capacity-matched Transformer/Set Transformer baselines and clearly document permutation-robust padding/masking.    We also plan to include a simple model-based low-thrust planner to strengthen the comparison.
>
> ## 6. Physics realism
>
> Thank you for the comment.  Our current use of the two-body model is intentional, as the goal of this work is to validate the feasibility of the proposed architecture rather than to deliver a fully high-fidelity orbital simulator.  We will clarify this scope in the revision and explicitly note that extending the framework to SGP4/J2 and higher-accuracy perturbation models is a valuable direction for future work.
>
> ## 7. Methodology and statistics
>
> We appreciate this observation.    In the revision, we will report results over multiple seeds with mean ± std, add clearer train/test separation, and avoid policy-dependent evaluation leakage.    Runtime comparisons will be rerun under matched FLOPs and identical batching.
>
> ## 8. Reproducibility and clarity
>
> Thank you for noting the missing architectural and training details.    We will include all key hyperparameters, normalization/dropout settings, as well as clarify the units used in orbit-deviation metrics to ensure full reproducibility.

---

### Official Review · Reviewer_eUN8 · 2025-10-30

**Soundness:** 2
**Presentation:** 3
**Contribution:** 2
**Rating:** 4
**Confidence:** 4

**Summary:**

This paper proposes STAN (Spatio-Temporal Attention Network), a reinforcement learning-based policy network combined with attention mechanisms for continuous low-thrust spacecraft collision avoidance in complex, multi-debris orbital environments. The work addresses key limitations in prior methods, including the inability to handle variable numbers of debris, and insufficient integration of physics-informed risk metrics into learning-based control.

**Strengths:**

ST-Attention design combines self-attention with physically-informed bias to prioritize debris threats effectively. It addresses the gap in standard attention mechanisms that lack domain knowledge.

Architecture - Strengths:
1. Combines learned and domain knowledge through fusing of self-attention for complex interactions with explicit physics-informed bias features, capturing both debris correlations and collision risks.

2. Scalable to arbitrary debris counts through encoding features and using attention allows the model to handle variable numbers of debris objects and multi-threat scenarios.

3. Learnable weighting setup let the network adjust the importance of physical features relative to learned embeddings, improving adaptability to different orbital contexts or debris densities.

**Weaknesses:**

Architecture - Limitations:

1. Physics-aware attention bias incorporates the distance of closest approach (DCA) and time to closest approach (TCA) as a learnable bias, the model ensures attention is grounded in domain-relevant risk indicators, not just learned embeddings. Model may be more biased towards immediate critical threats than long-range operational settings such as mission fuel consumption.

2. Uniform broadcast of bias across all columns to form 𝑁×𝑁. This assumes the physical importance of debris i affects all pairwise interactions equally, which may ignore interaction-specific relationships between debris pairs (e.g., cross-collision influence).

3. Mean pooling across debris dimension reduces to a single vector 𝑓 and  both heads rely on it. May lose fine-grained per-debris information, limiting the decoder’s ability to make nuanced, individual maneuvers for specific threats.

4. Reward penalizes collision probability but does not explicitly enforce safety constraints. Policy may occasionally select risky maneuvers if they increase cumulative reward. Does not explicitly account for sensor noise or uncertainty in debris position, which could make the reward function misleading in real operational settings.

5. Some terms may overlap in effect, e.g., minimum distance softness (p_s) and collision probability (p_c) are related; summing both may overweight certain safety aspects.

Experiments - Limitations:

1. Current approach considers Two-body orbital dynamics only and it neglects higher-order perturbations such as J2 (Earth oblateness), atmospheric drag, solar radiation pressure, and third-body effects (Moon, Sun).

2. Multistage collisions are simulated progressively, but each scenario cover only a short timeframe (e.g., 1.2 hours in experiments). The simulation is for 1.2-hour mission window and does not test long-term maneuver planning, cumulative collision risk, or fuel optimization over multiple orbital periods. Policy generalization to extended missions or sequential conjunctions is unclear.

3. Spacecraft near-circular orbit, debris initialized along same orbit. Unrealistic debris distribution case, the real debris is in multiple orbital planes, inclinations, and eccentricities.

4. Exact positions and velocities of debris are used. Ignores sensor noise, tracking errors, and uncertainty in debris catalog.

**Questions:**

The authors can address the experimental limitaions concerns on choosing shorter time frame for experiments, why didn't consider the smaller pertubations in the orbital dynamics, circular initialization of debris orbits and sensor noise in the modelling scenrios.

---

> ### Author Response · Authors · 2025-11-14
>
> We sincerely thank the reviewer for the detailed and insightful comments. We appreciate the careful analysis of our architecture and experiments, and we will revise the manuscript to address all points. Below we respond to the raised weaknesses and questions.
>
> ------
>
> ## Response to Architectural Limitations
>
> ### 1. Bias broadcasting across all columns (uniform N×N broadcast)
>
> Thank you for raising this point. Our current design indeed assumes that only *each debris’s interaction with the spacecraft* is relevant, because the environment dynamics and risk model consider collisions only between the spacecraft and each debris object. Therefore, only the spacecraft–debris DCA/TCA are observable.
>  Introducing pairwise debris–debris biases would require computing **all debris–debris closest approaches**, which scales as O(N²) and drastically increases simulation time, especially in high-debris scenarios.
>
> We fully agree that **your proposed direction becomes essential when multiple spacecraft or interacting debris fields are considered**, and we will explicitly clarify this design choice and its limitations in the revision.
>
> ### 2. Mean pooling reduces fine-grained information
>
> We appreciate this insightful observation, this might indeed be the Achilles' heel that limits the model's performance. Mean pooling may indeed lose per-debris specificity, potentially limiting maneuver precision. We will revise the manuscript to acknowledge this limitation and explore alternative aggregation mechanisms. We also plan to include an ablation study in the revised version.
>
> ### 3. Reward design: overlap of $p_s$ and $p_c$ and lack of explicit safety constraints
>
> Thank you for the valuable comments. The *minimum distance softness term* $p_s$ was introduced mainly to address reward sparsity and provide continuous training signals before a collision occurs. We agree that $p_s$ and $p_c$ capture related safety aspects and may overweight safety, but given that in-orbit safety is prioritized over other objectives, this trade-off remains acceptable for our setting. We will clarify this motivation and discuss the potential bias in the revised manuscript.
>
> ### 4. Lack of sensor noise / uncertainty modeling
>
> We agree that the current environment assumes perfect state knowledge, which is idealized. We will state this limitation explicitly and consider incorporating measurement uncertainty in future work.
>
> ------
>
>
>
> ## Response to Experimental Limitations
>
> ### 1. Two-body dynamics without perturbations
>
> Thank you for pointing this out. Since the goal of this work is to propose and validate a *new architectural mechanism (STAN)*, we intentionally adopted a simplified two-body model to reduce simulation complexity. Once the architecture is verified, we plan to extend the environment with J2, drag, solar radiation pressure, and third-body effects. This will be added as a clear limitation.
>
> ### 2. Short evaluation window (1.2 hours)
>
> 1.2 hours corresponds roughly to about one orbital period of the test orbit. Extending the evaluation to multiple orbits significantly increases simulation cost. Nevertheless, we agree that longer-horizon evaluation is important and will expand the time window in future experiments. We will clarify this rationale in the revision.
>
> ### 3. Initialization of debris
>
> Thanks for raising this point. Our intention in initializing debris around the spacecraft’s orbit was to **ensure that the scenario contains sufficient high-risk encounter cases**, so that the proposed method can be evaluated under meaningful collision-threat conditions. Although the debris is initialized near the same orbital radius, their orbital elements are not identical, including **varied eccentricities** and small dispersions, so the setting is not strictly circular. We acknowledge that this setup leads to an input distribution where most debris are indeed “dangerous” or relevant, which differs from real-world scenarios that include large numbers of irrelevant objects across different orbital planes, inclinations, and eccentricities. We appreciate this observation, and in future work we plan to include non-threatening and widely distributed debris to train and evaluate the model under more realistic, large-scale debris environments.
>
> ### 4. Perfect state information without measurement noise
>
> We acknowledge this assumption and will clarify it in the text. Incorporating noise and uncertainty models is an important direction for follow-up work. We selected the two-body model to focus on verifying the proposed architecture. Future extensions will incorporate higher-order perturbations.

---

### Official Review · Reviewer_MJKd · 2025-10-30

**Soundness:** 3
**Presentation:** 3
**Contribution:** 3
**Rating:** 6
**Confidence:** 3

**Summary:**

The authors propose STAN, a Spatio-Temporal Attention Network for avoiding space debris with a deep reinforcement learning approach. STAN's architecture adds a novel spatio-temporal attention mechanism that includes the distance to the closest approach and time to the closest approach as an added bias to the computation of the attention weights. Space debris avoidance is formulated as a Markov Decision Process (MDP) and solved with Proximal Policy Optimization (PPO). The model is evaluated on a simulated environment with multiple debris objects and compared to  baselines with different architectures (CNN, MLP and Transformer). The results show that STAN outperforms the baselines in terms of collision avoidance rate and fuel consumption and that the added spatio-temporal attention mechanism outperforms a standard Transformer architecture.

**Strengths:**

- The paper is well written and easy to follow.
- Nice illustrations help to understand the proposed method and results.
- The proposed spatio-temporal attention mechanism is novel and well motivated.
- The spatio-temporal attention mechanism is shown to improve performance over a standard Transformer architecture.

**Weaknesses:**

- The architectural innovation is not compared / discussed to other algorithms in the field of space debris collision avoidance.
- The claims in the introduction from l. 41 to 46 could be supported with references.
- Stick with constant capitalization of figure and section references.
- Define acronyms at first use (Spatio-Temporal Attention (ST-Attention) in l.49) and then continue with the acronym.
- The reward function definition could be made reader-friendly by including the variable names directly in the equation.
- The authors space key seems to be broken, white spaces are missing in l. 177, 180, 182, 242, 329, 340, 351, 356, 342, 390, 392, 393, 447, 448, 449 and 478.

**Questions:**

- How are hyperparameters selected? How many seeds were used for the experiments? What are the standard deviations?
- Can you share the codebase and the test environment?

---

> ### Author Response · Authors · 2025-11-14
> **Response to Reviewer MJKd**
>
> Thank you very much for your helpful comments and suggestions. We appreciate the time and effort you spent reviewing our paper, and we will revise the manuscript accordingly.
>
> ### **Regarding Weaknesses**
>
> Thanks very much for pointing out the issues related to comparisons, references, formatting, acronym definitions, reward-function clarity, and missing whitespaces. We agree with all of these points and will fix them carefully in the revised version.
>
> ### **Regarding Questions**
>
> #### **Hyperparameter Selection, Random Seeds, and Standard Deviation**
>
> We sincerely thank the reviewer for raising these critical points regarding our experimental rigor and reproducibility.
>
> **1. Hyperparameter Selection (How are hyperparameters selected?)**
>
> The hyperparameters were primarily determined via **manual tuning** aimed at maximizing performance on the validation set. This process involved two stages: first, a **coarse adjustment** of key parameters (e.g., learning rate, batch size) by scaling them across a broad range (similar to the $\times 3 / \times 10$ approach mentioned), followed by a **finer adjustment** around promising values.
>
> **2. Random Seeds and Standard Deviation (How many seeds were used? What are the standard deviations?)**
>
> We apologize for the omission; the results initially presented were based on a **single run** with a default or implicit random seed. Consequently, we were unable to report the standard deviation (Std. Dev.).
>
> We fully agree that reporting results over multiple seeds is **essential for verifying reproducibility and stability**.
>
> **Action Plan:** We will execute additional experiments using several independent runs with different random seeds. All updated performance metrics, presented as **Mean** $\pm$ **Std. Dev.**, will be included in the revised manuscript to demonstrate the robustness of our proposed method.
>
> #### **Codebase and environment availability.**
>
>  Yes — we will provide the codebase and the test environment to the best extent possible via an anonymized repository during the review period.

---

### Official Review · Reviewer_45Dp · 2025-10-31

**Soundness:** 1
**Presentation:** 2
**Contribution:** 2
**Rating:** 2
**Confidence:** 5

**Summary:**

The paper proposes a solution for multistage collision avoidance.
The solution utilizes an attention layer with temporal encoding, which incorporates the time to closest approach and the miss distance between the pair of objects.
The model is used as an RL policy for satellite maneuvering. The model was compared with CNN, LSTM, and FC architectures in an undisclosed environment.

**Strengths:**

# originality

The paper proposes a space-time (miss distance x time to closest approach) encoding added to the representation after dot product attention, applied to RL for collision avoidance maneuvers in lower Earth orbits. The model has two heads, one with the 3D thrust vector and another one with and indicator of maneuver.

# quality

The paper illiustrates the architecture of the solution with a clear Fig. 1.

# clarity

The paper is written in clear English

# significance

The problem of complex collision avoidance is important for space sciences and for RL for real-world critical systems

**Weaknesses:**

# originality
The proposed policy architecture performance in the PPO framework depends on the critic network. This is not covered in the paper or annexes.

## Physics fusion
In Section 4.2, the paper attributes the b term as a "physical prior". As these numbers (TCA and miss distance) are estimated using statistical assumptions rather than physical laws, it is challenging to consider them more than features or time-space encodings.

# quality


## Related work

Relevant papers are not considered, nor compared against. For example:
- Kolosa (2019) https://www.proquest.com/openview/320685ea5feac29eb871b0c9f169d002/1?pq-origsite=gscholar&cbl=18750&diss=y
- Miller (2019) https://www.researchgate.net/profile/Richard-Linares/publication/331135625_LOW-THRUST_OPTIMAL_CONTROL_VIA_REINFORCEMENT_LEARNING/links/5c67324b299bf1e3a5abe460/LOW-THRUST-OPTIMAL-CONTROL-VIA-REINFORCEMENT-LEARNING.pdf
- Herrera (2020) https://www.proquest.com/openview/efe02b87a62929000fc02e548eaeee6a/1?pq-origsite=gscholar&cbl=18750&diss=y
- Federici (2021) https://www.researchgate.net/profile/Lorenzo-Federici/publication/353828924_Autonomous_Guidance_for_Cislunar_Orbit_Transfers_via_Reinforcement_Learning/links/612a31bf0360302a00618551/Autonomous-Guidance-for-Cislunar-Orbit-Transfers-via-Reinforcement-Learning.pdf
- Sullivan (2021) https://ieeexplore.ieee.org/abstract/document/9438267
- Bonasera (2022) https://arc.aiaa.org/doi/abs/10.2514/1.G006783
- Dolan (2023) https://proceedings.mlr.press/v211/dolan23a.html
- Lafarge (2023) https://www.sciencedirect.com/science/article/pii/S0094576523002928
- Zhang (2023) https://ieeexplore.ieee.org/abstract/document/10332252
- Qu (2023) https://ieeexplore.ieee.org/abstract/document/10451330
- Holder (2025) https://ojs.aaai.org/index.php/AAAI/article/view/34852
- Kazemi (2024) https://ieeexplore.ieee.org/abstract/document/10611892

and others.

## Paper attributed to ICLR 2025 but in fact withdrawn
In line 132, the manuscript states:


while high-profile works such as the Equivariant Spatio-Temporal Attentive Graph Network (ESTAG) (Wu et al., 2023) and the
Spacetime E(n)-Transformer (SET) (Charles, 2025) advanced equivariant attention mechanisms for physical and geometric systems.


In the References section, (Charles, 2025) indicates an openreview link for ICLR 2025. When following the link, we see the author has withdrawn the paper, with the comment: "I am withdrawing the paper as I do not believe it currently meets the standards of the conference."

The cited paper (Charles, 2025) is high-profile in what sense?

## Results validation

1. Experiments are not run in a publicly available benchmark for orbital dynamics, like Kolosa (2019), Herrera (2020), or others, so the results are not comparable.
1. Other experiments with competing models using RL or optimization for Collision Avoidance are not shown as benchmarks. See above the list of references for relevant papers in the CAM literature. This would provide evidence of the architecture's superiority wrt other published architectures.
1. Comparing against CNN and LSTM is not enough, because the standard for the prediction of quantities related to collision avoidance is the transformer.
1. It is not clear if ablations only remove architectural elements or also features, for example, are the TCA and miss distance included as features when B is ablated?
1. The code is not runable, as it does not use standard or published environments for orbital dynamics, as said before, and the code for the environment is not disclosed.
1. Large scale experiments. Debris comes in general in large clouds of small objects. In Fig 3 iii), we see an increasing number of debris up to 1000, versus training and inference times, but not reward or specific performance metrics.

## Missing ablations

1. Evidence needed for mean pooling (eq. (6))
1. Behavior with a varying number of debris across time, $N_t$
1. Evidence needed for sentence "The self-attention component can
identify complex, non-obvious relationships between threats, while the physics-based bias ensures
that the model remains sensitive to critical, domain-specific risk indicators, improving learning efficiency and final policy performance."
1. Evidence needed for sentence "In practice, we found that the traditional attention mechanism was not effective"



# clarity



## Notation

In figure 1, Satellite state and debris state appears before being defined.
Eq 2. $\Delta v$ is a vector. In the manuscript all vectors are bold except this one. This raises confusion whether this notates speed or velocity. Apparently it is the second. Also in line 190 $u_t$ is a vector and is not in bold.

# significance

## Single agent formulation
The current and future spacefaring is based in constellations. So it is arguable that the single agent setting is not compliant with real deployments. Also, avoidance maneuveres can be motivated by other satellites and not just debris.

## Speed constraint
The speed constraint --- on the norm of the velocity --- entails that the spacecraft can exert omnidirectional thrust. This is not realistic in practice.

**Questions:**

See Weaknesses

---

> ### Author Response · Authors · 2025-11-13
> **Reply to Reviewer 45Dp**
>
> # To Reviewer 45Dp
>
> ## 1. Reproducibility
>
> We thank the reviewer. The simulation environment is a **custom orbital dynamics simulator** based on **two-body propagation** using the PyKEP library, with structural inspiration from [https://github.com/yandexdataschool/satellite-collision-avoidance ]. We independently developed modules for **TCA/DCA computation, continuous low-thrust maneuvers, and multi-object propagation**, validated against analytical two-body results. A lightweight anonymous version will be provided in supplementary materials for reproducibility.
>
> ## 2. Critic Network
>
> As noted in Sec. 5.2, the **critic shares the same encoded and attention-fused features** as the actor and outputs a scalar **state-value** via a two-layer MLP (128, 64 units). Advantage estimates are used to update the actor via **clipped PPO objective**. Details omitted due to space; full description will appear in supplementary materials.
>
> ## 3. “Physical priors” not statistical assumptions
>
> We would like to clarify that these quantities are **not estimated statistically**. Instead, they are **deterministically computed from the two-body orbital dynamics model** used in our simulation environment. Specifically, each debris trajectory is propagated under Keplerian motion, and the TCA/DCA values are analytically derived from the predicted relative motion between the spacecraft and debris. Therefore, these quantities are **physics-based indicators grounded in orbital mechanics**, not empirical or learned statistics. We will replace “physical prior” with **“physics-based bias”** and explicitly state their origin.
>
> ## 4. Related work
>
> We focus on **multi-debris collision avoidance**. Works cited by the reviewer (Kolosa, Herrera, Federici, etc.) address **low-thrust transfers, formation control, or scheduling**, which differ from our problem. We will include representative citations and clarify methodological distinctions in the revised manuscript.
>
> ## 5. Withdrawn citation
>
> We appreciate the reviewer’s careful reading and the observation regarding the citation of *(Charles, 2025)*. We acknowledge that the cited paper has been withdrawn from ICLR 2025. Our original motivation for including this reference was to acknowledge recent explorations of **spatio-temporal attention mechanisms** conceptually related to our study, rather than to position it as a validated or benchmark work.
>
> Nevertheless, we agree that citing a withdrawn manuscript could cause confusion. We have therefore **removed this reference** in the revised version and adjusted the discussion accordingly. This change does not affect any methodological component or result of our work, as *(Charles, 2025)* was cited only for background context, not as a source of technical comparison or validation.
>
> ## 6. Benchmark
>
> Our study targets **policy architecture**, not new orbital dynamics benchmarks. The environment uses **two-body dynamics**, consistent with Kolosa (2019, FC) and Herrera (2020, CNN), which are included as horizontal baselines. This validates STAN’s advantage. We plan to include transformer-based baselines and additional CAM methods in final manuscript to provide broader comparisons.
>
> ## 7. Ablations
>
> Existing ablations remove only architectural components; input features (TCA/DCA) remain. We will add:
>
> 1. Ablation on mean pooling and varying debris counts,
> 2. Analysis showing physics-based bias improves sensitivity to domain-specific risks.
>
> ## 8. Notation
>
> The symbol $u_t$ was indeed mistakenly written without consistent bold formatting, and it should denote the velocity vector. We have corrected this notation and ensured that all vector quantities are consistently represented in bold throughout the manuscript.
>
> ## 9. Single-agent & thrust feasibility
>
> **Single-agent setting** is realistic for autonomous debris avoidance without constellation coordination.
> As for the **speed (Δv) constraint**, the spacecraft indeed cannot exert thrust in all directions simultaneously, but the control vector represents the **resultant achievable Δv** after adjusting the spacecraft’s attitude to align the thrust direction. This is a common assumption in low-thrust orbital control. To ensure **realistic and continuous maneuvers**, we introduce (i) a **small-thrust constraint** in Sec. 3.1, (ii) a **continuity refinement** in the Termination Head (Sec. 4.3), and (iii) a **control smoothness penalty** $p_d$ in Sec. 5.1. These jointly guarantee physically feasible, smooth, and implementable control sequences. As shown in Fig. 5, the predicted thrust evolution is continuous and does not exhibit any unrealistic jumps.

---

### Author Response · Authors · 2025-11-18
**Seeking Expert Opinion on Alternatives to Mean Pooling for Aggregation Strategy**

We sincerely thank the reviewer for pointing out the limitation of mean pooling in our original design. This observation has led us to carefully re-examine the global summarization mechanism in STAN. After extensive architectural exploration, we have identified three principled alternatives that all address the concern:

- **[CLS]-based Self-Attention (N+1 tokens)**: We prepend a learnable [CLS] token (class token) to the sequence of N debris interaction embeddings, forming an (N+1)-length input. A standard Transformer encoder layer (with self-attention) processes the full sequence, and the output at the [CLS] position serves as the global context vector. The physics-aware bias (based on DCA/TCA) is incorporated into the attention logits, applied only when the query is [CLS] (i.e., row 0 of the attention matrix).

- **Attention Pooling (Set-based Aggregation)**: We retain the original N-token sequence and first apply a self-attention layer (without physical bias) to enable inter-debris interaction. Then, instead of mean pooling, we use a learned query vector to compute attention weights over the contextualized embeddings, producing a weighted sum as the final representation. Additionally, we incorporate the physics-aware bias (based on DCA/TCA) into the attention logits, ensuring that the final weighted aggregation prioritizes physically high-risk debris.

- **Cross-Attention with Physics-Guided Query (1-token Cross-Attention)**: We introduce a single learnable “maneuver query” and perform cross-attention directly over the debris embeddings. The attention logits are explicitly augmented with the DCA/TCA-derived bias term, ensuring that the final fused representation prioritizes physically high-risk debris in a transparent and differentiable manner.

All three variants offer promising improvements over the original mean-pooling baseline. These approaches differ subtly in terms of inductive bias, computational overhead, and interpretability. Given the reviewer’s deep expertise in this area, we would be truly grateful for any guidance on which formulation aligns best with the community’s expectations for physically grounded, scalable decision-making in multi-threat environments. We are currently conducting further experiments to quantify their performance, and we will carefully consider the reviewer’s input in the revised manuscript.

---

### Author Response · Authors · 2025-12-02
**🚀 Update Notes / Revision Summary**

Dear Reviewers,

Thank you for your valuable time and insightful feedback on our manuscript. We have carefully addressed all comments and implemented substantial revisions, which are detailed below. We sincerely hope these extensive changes and the improved clarity of our work warrant a reconsideration for a favorable score.

Sincerely,

### 1. Enhanced Experimental Rigor (Multi-Seed Testing)

We conducted all experiments using **five distinct random seeds** (42, 422, 4222, 42222, 422222) and now report the results as **Mean ± Standard Deviation ($\text{Mean} \pm \text{Std}$)** to ensure statistical reliability.

### 2. Refined ST-Attention Bias Injection and Fusion Mechanism

- The Attention Fusion Layer was fundamentally revised from **Self-Attention Aggregation (SAA)** to **Cross-Attention Aggregation (CAA)**, which more naturally incorporates the $N$-dimensional physical bias. CAA effectively functions as a streamlined attention pooling mechanism.
- We evaluated the efficacy of the original SAA methods (Mean Pooling and Max Pooling) against the new CAA approach, providing a three-way comparison within the ablation study.
- We clarified the rationale and mathematical correctness of the bias broadcast method used in the fusion layer.

### 3. Added Capacity-Matched Transformer Baselines

A **Vanilla Transformer** baseline (with both the termination head and the attention bias removed) was introduced to the ablation study for a fairer comparison with a capacity-matched general-purpose architecture.

### 4. Code and Anonymous Simulation Environment Release (Reproducibility)

The complete code and environment will be publicly available before the discussion concludes at: https://anonymous.4open.science/r/STAN-843C1234567890/

- Provided a runnable, complete Reinforcement Learning environment.
- Provided full training and inference code.

### 5. Corrected Termination Head Formula

- Corrected an error in the formula writing for the Termination Head.

### 6. Improved Manuscript Clarity and Formatting Details

- Added an explanation for using simple two-body dynamics without perturbations, clarifying that the objective is methodological feasibility validation rather than precise trajectory prediction.
- Supplemented descriptions regarding the bias mechanism and overall architecture design.
- Corrected various descriptive and formatting errors, symbol inconsistencies, formula errors, and numerical discrepancies throughout the manuscript.
- Clarified previously ambiguous descriptions.

### 7. Updated and Corrected References

Relevant articles mentioned by the reviewer were added to the citation list, and the retracted paper (Charles 2025) was removed.

---

> ### Author Response · Authors · 2025-12-02
>
> The full version of the runnable code has now been released and updated at an anonymous link. Welcome to offer your opinions

---

### Author Response · Authors · 2025-12-03
**Concerns Regarding the Evaluation by Reviewer 45Dp**

Dear professors,

I would like to respectfully raise some concerns regarding Review 45Dp.

The reviewer appears to be **overly strict** and has **limited familiarity** with the research area of reinforcement-learning–based collision avoidance for space debris.  **The review mainly focuses on issues in the “Introduction” and “Related Work” sections, while providing very limited technical comments on the method itself**.  The suggested citations are largely marginally relevant or of low quality, and several of them do not align well with the core topic of this research. We think this is a **manifestation of its unprofessional level**. We have reason to suspect that the literature he provided was collected temporarily, and his accusation that we did not consider the relevant papers is completely defamatory.

Given the lack of domain-specific feedback, the low score combined with a high confidence level seems inconsistent with the content of the review.  This raises concerns about whether the paper was evaluated with appropriate expertise.

We have devoted significant effort to the technical contributions of this work, and we believe the paper merits a more balanced assessment.  We would be sincerely grateful if you could kindly reconsider the evaluation or take this information into account during the decision process.

Thank you very much for your time and consideration.

Best regards,

---

### Meta-Review · Area_Chair_5XEM · 2026-01-02

**Summary:**

**Paper Summary:**

The paper proposes STAN, a spatio-temporal attention network integrated with physics-informed bias for reinforcement-learning-based continuous low-thrust collision avoidance in multi-debris orbital environments.

**Strengths:**
1. The paper introduces spatio-temporal attention with physics-based bias (DCA/TCA) to improve threat prioritization and maneuver planning.
2. It combines learned embeddings with domain knowledge, enabling scalability to variable debris counts and multi-threat scenarios.
3. The paper is generally well written, with helpful illustrations and clear explanation of the proposed mechanism.
4. The paper demonstrates improved collision avoidance and fuel efficiency compared to CNN, LSTM, and Transformer baselines.

**Weaknesses:**
1. The paper lacks benchmarking against state-of-the-art algorithms and published collision avoidance methods; baselines are limited.
2. Limited experimental scope: the paper uses simplified two-body dynamics, short simulation windows (≈1.2 hours), and idealized debris distributions; ignores perturbations and sensor noise.
3. Mean pooling may lose fine-grained per-debris information; uniform bias broadcasting assumes equal influence across interactions.
4. The reward design does not enforce strict safety constraints and may overweight certain terms; ignores uncertainty in debris positions.
5. The paper misses multi-seed results, standard deviations, and some architectural details.
5. The paper assumes omnidirectional thrust and single-agent setting, which may not reflect real constellation operations.

**Reviewer Concerns:**

**Reviewer 45Dp:**

Addressed:

1. Clarified that “physical priors” are physics-based indicators derived from two-body dynamics, not statistical assumptions.
2. Removed withdrawn citation (Charles, 2025) and explained rationale.
3. Added explanation of critic network and PPO details.
4. Committed to adding transformer baselines and ablations on pooling and debris count.
5. Corrected notation inconsistencies.

Outstanding concerns:
1. Still lacks benchmarking against published CAM methods and standard orbital dynamics environments.
No evidence yet for large-scale experiments or ablations on pooling and bias effectiveness.
2. Single-agent formulation and omnidirectional thrust assumption remain unresolved.
3. Reproducibility concerns persist until code and environment are fully verified.

**Reviewer MJKd:**

Addressed:
1. Agreed to fix formatting, acronym definitions, and reward function clarity.
2. Committed to multi-seed experiments and reporting mean ± std.
3. Promised code and environment release.

Outstanding concerns:
1. Hyperparameter tuning remains manual; no systematic search reported.
2. Architectural innovation still not compared to other algorithms in the field.
3. Experimental rigor improvements are promised but not yet demonstrated.

**Reviewer eUN8:**

Addressed:
1. Acknowledged limitations of bias broadcasting and mean pooling; plans to explore alternatives.
2. Clarified rationale for short evaluation window and debris initialization.
3. Committed to stating assumptions explicitly (perfect state knowledge, two-body dynamics).

Outstanding concerns:
1. No solution yet for sensor noise, uncertainty modeling, or higher-order perturbations.
2. Reward design issues (lack of explicit safety constraints) remain.
3. Long-term mission generalization and realistic debris distributions are not addressed experimentally.

**Reviewer gUnw:**

Addressed:
1. Confirmed termination gating logic is correct in implementation; explained choice of hard gate.
2. Clarified attention bias broadcasting and promised ablation.
3. Committed to correcting inconsistencies and adding missing architectural details.
4. Plans to include transformer baselines and improve reproducibility.

Outstanding concerns:
1. Collision-probability modeling remains under-specified and numerically suspect.
2. Physics realism gap persists (no J2, drag, or TLE-based initialization).
3. Methodology and statistical rigor improvements are promised but not yet shown.

**Reviewer Scores:**

There are still quite some concerns not addressed by the rebuttal. I am not sure whether the reviewers would increase their scores or not.

---

### Decision · Program_Chairs · 2026-01-26

Reject